# Continuous Production of Water-Borne Polyurethanes: A Review

**DOI:** 10.3390/polym12122875

**Published:** 2020-11-30

**Authors:** Xuehua Liu, Wei Hong, Xudong Chen

**Affiliations:** Key Lab Polymer Composite & Key Laboratory for Polymeric Composite and Functional Materials of Ministry of Education, Guangdong Engineering Technology Resarch Center for High-Performance Organic, School of Chemistry, Sun Yet-Sen University, Guangzhou 510275, China; aliuxuehua@126.com (X.L.); hongwei9@mail.sysu.edu.cn (W.H.)

**Keywords:** water-borne polyurethane, continuous production, reaction mechanism, viscosity

## Abstract

Water-borne polyurethanes are novel functional polymers that use water as the dispersion medium. When compared with solvent-borne polyurethanes, water-borne polyurethanes are more environmentally friendly and easier to transport and store. Water-borne polyurethanes have attracted increasing attention due to their extensive applications in plastics, paints, adhesives, inks, biomaterials, and other fields. In this study, the characteristics of water-borne polyurethanes were discussed, followed by a review of studies detailing reaction procedures and mechanisms for their continuous production. Additionally, current and future applications of continuous production processes for water-borne polyurethanes are presented.

## 1. Introduction

Water-borne polyurethanes are binary colloid water-soluble polymers that are obtained by introducing hydrophilic groups (carboxylate, sulfonate, quaternary ammonium salt, or hydrophilic segments) into polyurethane molecular chains in order to render it having a hydrophilic property. Compared with the traditional organic solvent polyurethane system, the water-borne polyurethanes exhibit environmentally-friendly, safe and reliable, excellent compatibility and modifiability.

Water-borne polyurethanes have an excellent film-forming capacity, high viscosity, low temperature resistance, and low cost [1]. With the environmental protection consciousness increasing, water-borne polyurethanes have been widely applied in paints, printing inks, and adhesives in order to replace solvent-borne polyurethanes [2]. Batch processes mainly synthesize water-borne polyurethanes, which time and energy intensive, have a low solid contents, high viscosity, and hard to control performance. The continuous production of water-borne polyurethanes uses water as a solvent or only contains a small amount of organic solvent, which is more environmentally friendly. The produced water-borne polyurethanes have controllable viscosities and hydrophilicity. Additionally, continuous production is suitable for large-scale production, and the products have enhanced performance stability [3]. Herein, several continuous synthesis processes of water-borne polyurethanes are discussed.

Water-borne polyurethane dispersions are environmentally-friendly polymers that can be readily modified by mixing with other resins. They are characterized by simple formulas and low cost, and, thus, exhibit great application potential [4]. It has been widely applied in industry and daily life, due to their tunable soft and hard segments, good low-temperature resistance, high flexibility, and strong adhesion [5].

Water-borne polyurethanes can be applied as construction and automotive paints, leather industry, and adhesives based on their low toxicity, inflammability, good wear and corrosion resistance, and strong adhesion [6,7]. The batch-fed production process presents problems of small production scale, labor-intensiveness, and would significantly suffer from human factors, which affect the quality of water-borne polyurethanes and the resulting dispersions have a short shelf life. In contrast, the continuous production of water-borne polyurethanes provides a highly automated method that is suitable for large-scale production and it endows the final product with enhanced performance stability. As one of the major water-borne polyurethane production technologies, continuous production technology has been involved in previous reviews [8], but it has not been summarized as a special review to be best of our knowledge. This review presents a recent break-through advance in continuously produced water-borne polyurethane and discusses their applications in adhesive, coating, and sewage disposal. This review demonstrates the effectiveness of continuous product of water-borne polyurethane and, in particular, the structural characteristics and reaction mechanism, continuous synthesis methods, and the applications of water-borne polyurethanes. All of these would provide scientific researchers greater convenience in researching water-borne polyurethanes via continuous production.

## 2. Synthesis and Main Properties of Waterborne Polyurethanes

### 2.1. Classfication

Water-borne polyurethanes can be classified as aqueous solutions, dispersions, and emulsions, according to their appearance and particle size. Aqueous solutions are colorless and transparent with particle sizes of less than 1 nm; dispersions are slightly turbid, white, and translucent, with particle sizes ranging from 1 to 100 nm; emulsions are white and turbid, with particle sizes larger than 100 nm.

Water-borne polyurethanes can be classified as single-component or two-component systems, depending on their application strategy. Single-component water-borne polyurethanes can be used without crosslinkers in order to achieve desired properties, while two-component water-borne polyurethanes generally required crosslinkers.

Water-borne polyurethanes can also be classified as cationic, anionic, zwitterionic, or nonionic, according to the ionic groups on the molecular chain and their charge. Moreover, water-borne polyurethanes are mainly divided into polyols and polyisocyanates, according to their reaction raw materials. For example, polyols can be divided into polyesters, polyethers, and polyolefins, while polyisocyanates can be divided into aliphatic, aromatic, and cycloaliphatic.

### 2.2. Structure and Synthesis Mechanism

Water-borne polyurethane molecules have a block structure that consists of repeat units of hard and soft segments. The hard segment of water-borne polyurethane is formed by the reaction of isocyanates with polyols, hydrophilic chain extenders, which generate polar groups, such as carbamates or ureas. These polar groups have strong intermolecular bonding forces, which make some hard segments connect together in order to form a rod-like structure. The conformation does not easily change, which serves as a physical cross-linking point in polyurethane segments. At the same time, molecules have relatively strong bonding forces and can easily aggregate, which makes it easy to separate the soft segments [9,10]. Figure 1 shows the reaction process.

The soft segment is composed of polyether, polyester polyols, or low-molecular-weight polyol segments. Soft segments cross-link with hard segments and they account for most of the polyurethane molecular chain [11]. Polyether-type water-borne polyurethanes contain many flexible ether bonds, which have good hydrolysis resistance, moisture permeability, and flexibility. Polyester water-borne polyurethanes contain many strong polar ester bonds, which have poor hydrolysis resistance; however, they have good mechanical performance, weather resistance, antifungal properties, and excellent low-temperature elasticity.

The main raw materials involved in the synthesis of water-borne polyurethanes are diisocyanates or polyisocyanates and polyols (including polyether type, polyester type, and polyolefin type), which are connected in an alternating fashion in the molecular chain. During the initial stage of the reaction, diisocyanates and polyols undergo pre-polymerization in order to produce a prepolymer with a low molecular weight. The prepolymer is connected by chain extenders, diols, diamines, etc., to form a polymeric chain. In the polymer chain, the isocyanates and chain extenders form rigid hard segments, while the flexible polyols form soft segments; and, the hard and soft segments form block copolymers [12].

TDI (toluene diisocyanate), MDI (diphenylmethane diisocyanate), and PAPI (polyphenyl polymethyl polyisocyanate) are commonly used isocyanates. Among these, water-borne polyurethanes that are prepared with TDI have good mechanical performance, low boiling point, and high toxicity. MDI is prone to dimerization, but it is less toxic and can generally be stored at low temperatures. PAPI has a high molecular weight, high boiling point, and low toxicity. The prepared water-borne polyurethanes have a high crosslink density and good mechanical performance due to the large number of isocyanate groups in molecules [13,14,15].

Polyols include polyether polyols, polyester polyols, and castor oil. Among them, polyether polyols are typically formed by the ring-opening polymerization of monomers, such as ethylene oxide, epoxypropane, and tetrahydrofuran. Examples include dihydroxy polyethylene oxide ether, trihydroxy polyoxyethylene ether, and tetrahydroxy polyoxyethylene ether, using KOH as the catalyst. Polyester polyols are usually formed by the reaction of a dibasic acid and excess polyol or via the ring-opening reaction of lactones, common diols, or polyols, such as ethylene glycol, diethylene glycol, trimethylolpropane, glycerin, etc. Castor oil is a mixture of ricinoleic acid and glycerides.

Diols and diamines are used as chain extenders. Diols are aliphatic and aromatic diols with low molecular weights (e.g., ethylene glycol, 1, 4-butanediol, and hydroquinone hydroxyethyl ether), diamines are primarily aromatic amines (e.g., benzidine). Catalysts and other additives are often tertiary amines, organostannic compounds, and mixed catalysts, etc. Tertiary amines include triethylamine, triethnolamine, and diaminopropane. Organostannic compounds include dibutyltin dilaurate and stannous octoate. Mixed catalysts are mixtures of amines and organotin compounds, while foaming agents, stabilizers, and fillers have also been used as catalysts.

The reaction of diisocyanates and diols produces carbamates; the reaction of polyisocyanates with polyols, polyethers, or polyesters will form a polyurethane; Figure 2 shows the reaction process.

Lei et al. [16] used a prepolymer and a solvent-free synthesis route in order to synthesize a green water-borne polyurethane. The selected chain extender was sodium 2,4-diaminobenzene sulfonate with low toxicity. The reaction of the prepolymer with sodium 2,4-diaminobenzene sulfonate was carried out in a mild and controllable manner, due to the presence of NCO groups. Figure 3 shows the specific reaction process of this experiment. Polyethylene glycol-1000 (PEG-1000) and polytetramethyl ether glycol-2000 (PTMEG-2000) were dried at 120 °C for 3 h. PEG-1000, PTMEG-2000, and diphenylmethane diisocyanate-50 (MDI-50) were added to a three-necked flask equipped with a mechanical stirrer and a thermometer and reacted for 2 h at 80 °C under continuous stirring (200 rpm). A completely NCO-terminated hydrophilic water-borne polyurethane dispersion was obtained. In another work, a novel water-borne polyurethane dispersion was also prepared by the prepolymer process and in-situ water reaction. The results of differential scanning calorimetry and dynamic thermomechanical analysis demonstrated that the prepared polyurethane was amorphous with good thermal stability and with more hydrogen bonding in order to form physical crosslinks with plastic resistance.

The –N=C=O group of isocyanate has highly unsaturated free groups and it is highly reactive. The nucleophilic center of the hydrogen-containing molecule will attack the electropositive carbon atom, which results in nucleophilic addition polymerization. For example, the reaction mechanism between an isocyanate and alcohol in the absence of a catalyst is as follows:
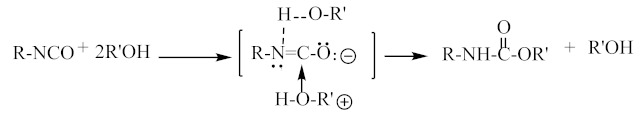
(1)

The reaction equation of an isocyanate and polyether/polyester polyol in the presence of a catalyst is as follows:
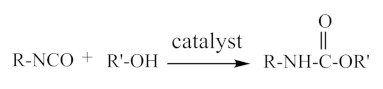
(2)

Urethane groups that are generated by the reaction contain many active sites. The isocyanate further polymerizes by heating or crosslinking in order to form a linear or nonlinear polyurethane.
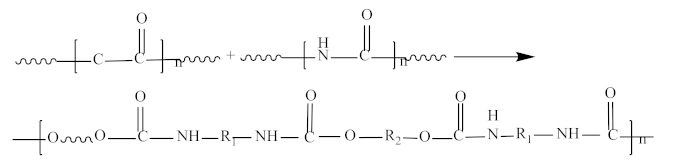
(3)

The properties of the resulting polyurethane can be modified by changing the type of polyol, the ratio of soft segments and hard segments, the types of chain extenders, the ratio of various functional chain extenders, and the order in which they are added [17]. Polyurethanes have highly designable structures, and they exhibit different properties, which makes them suitable for a wide range of applications in inks, elastomers, artificial leather, soft and hard foams, paints, and adhesives.

### 2.3. Characteristics of Water-Borne Polyurethanes

#### 2.3.1. Controllable Viscosity

The viscosity of water-borne polyurethanes is low and it can be adjusted by adding water-soluble thickening agents and water. In addition to external polymer thickening agents, ion charge, core-shell structure, and emulsion particle size also affect the viscosity [18]. The higher the density of free radicals in the main chain and side chains of water-borne polyurethanes, the greater the viscosity of the resulting polyurethane emulsion. Kim et al. [19] demonstrated that water-borne polyurethanes with ionic soft segments produced solutions with much lower viscosity, which resulted in a smaller particle size and larger dispersion viscosity. The molar ratio of NCO/OH affects the viscosity of water-borne polyurethanes. By increasing the molar ratio of NCO/OH, the content of polar bonds, such as carbamates and allophanates, increased, causing dispersed polymer particles to aggregate, which increased the dispersion particle size. Decreasing the dispersion particle size and increasing the polar groups restricted the activity and diffusivity of the polymer resin and reduced its viscosity. The solid content, molecular weight of the polyurethane resin, crosslinker, and other factors had little effect on the viscosity of water-borne polyurethanes, but they increased the molecular weight and branching degree of polyurethane and improved its cohesive strength [20,21].

#### 2.3.2. Biodegradability

Water-borne polyurethanes contain repeating carbamate segments in their main chain. The main chain usually consists of soft segments with a glass transition temperature that is lower than room temperature and a rigid segment with a glass transition temperature higher than room temperature. The soft segments are composed of oligomeric polyols (such as polyesters and polyethers), and the hard segments are composed of diisocyanates and low-molecular-weight chain extenders (such as diamines and glycols). The hard segments are highly polar and they have strong mutual attraction. The hard and soft segments tend to spontaneously separate; therefore, hard segments easily gather and form many micro-regions distributed in the soft segment phase, leading to microphase separation. The microphase surface structure of water-borne polyurethane is very similar to that of biofilms, which gives it good bio-compatibility. The chain structure of a water-borne polyurethane determines its biodegradability [22]. During synthesis, water-borne polyurethanes can be designed by selecting different blocks and adjusting the ratio between soft and hard segments in order to afford water-borne polyurethanes with different chemical structures (e.g., linear, branched, crosslinked), mechanical performances (rigid, flexible), and thermal stability to meet different application requirements.

#### 2.3.3. Curing via Intramolecular Interaction of Polar Groups

The main chain of polyurethanes contains carbamate repeat units that contain urethane bonds, which can form physical crosslinking points via hydrogen bonds between or within the polymer molecule, thus giving it excellent mechanical performance [23]. Water-borne polyurethanes do not contain –NCO groups, and most water-borne polyurethanes are mainly cured by cohesive and adhesive forces, such as hydrogen bonds and van der Waals forces that are generated by polar groups within the molecule.

#### 2.3.4. Controllable Thermal Stability

Water-borne polyurethanes are composed of hard and soft segments, with significantly different chemical properties. Their thermodynamic incompatibility to make them exhibit different glass transition temperatures. Water-borne polyurethane chains begin to thermally decompose with their hard segments, where the chemical composition, content, length, and structure affect their thermal stability. In the initial stage of thermal decomposition, the hard segments usually show an ordered or semi-crystalline structure, which can enhance the thermal stability of water-borne polyurethanes. Thus, changing the hard segment greatly affects the thermal stability.

The second stage of thermal degradation mainly involves soft segments. As the polyol content increases in soft segment, there are fewer side chains, which produces a more regular structure, and it improves the heat resistance of the resulting water-borne polyurethane [24]. Upon increasing the molar ratio of NCO/OH in polymers, the concentration of hard segments (urea) and hydrogen bonding increases, leading to a greater extent of crosslinking. As the crosslinking of the polymer increases, so does its stability increases; hence, the *T*_g_ increases with the NCO/OH molar ratio [25].

The greater variety of raw materials and structures that constitute the hard segments of polyurethanes, the more straightforward and effective in improving the thermal stability of polyurethanes by changing the hard segment. The soft segments of water-borne polyurethanes are generally composed of oligomeric polyesters or polyether polyols. When compared with ethers, esters have greater cohesive energy and can easily form hydrogen bonds with the amino group of urethanes; therefore, the heat resistance of polyester polyurethanes is better than that of polyethers polyurethane [26]. The length and molecular weight of soft segments also affect the thermal stability of polyurethanes—the longer the length, the higher the thermal stability. As the molecular weight increases, the thermal stability of polyurethanes can be enhanced in the initial thermal decomposition stage; however, it will decrease in the later stage [27]. In summary, water-borne polyurethanes with controllable thermal stability can be obtained by improving the materials and structures of hard and soft segments.

## 3. Research Progress in Continuous Production of Waterborne Polyurethane

The current processes for producing water-borne polyurethanes in industry and academia mainly include continuous, semi-continuous, and batch-fed processes. Although batch-fed processes are technologically simple, there are problems, such as poor product stability, high energy consumption, high labor intensity, and high cost. Semi-continuous and continuous processes have been implemented industrially due to environmental concerns. Conventional batch-fed processes for producing water-borne polyurethanes are not suitable for producing high-yield adhesives, making it difficult to control the particle size and distribution of the colloidal particles due to production equipment and human factors. The prepared products show significant differences, with the problems of low solid content and poor initial adhesion, which them unsuitable as electrolytes, heat-sensitive materials, and flame retardants. Moreover, organic solvents are involved in the production process, which are difficult to recycle, and organic solvent bubbles may be mixed into the product, which are difficult to eliminate.

In 1962, Bayer first introduced hydrophilic groups in order to increase the self-dispersibility of molecules in water. This self-emulsification method improved the storage stability of emulsions, and also improved the water and solvent resistance of the resulting films, which solved problems with the external emulsification method [28]. In 1969, Bayer synthesized a polyurethane water dispersion without emulsifiers, which led to the development of a method in order to prepare water-borne polyurethanes. Subsequently, water-borne polyurethanes developed rapidly and they were used in many fields, such as leather finishing, gloves, catheters, and paints [29]. Since the 1980s, Bayer and other companies have issued many patents on the continuous production of water-borne polyurethanes, which solves the limitations and shortcomings of batch-fed production methods that rely on high-shear agitators with high energy consumption. Additionally, continuous production only requires short reaction times and produces stable water-borne polyurethane dispersions [30].

## 4. Continuous Production Processes

The continuous production of water-borne polyurethane provides improved the automation, stable product performance, scalable production scale, and easy maintenance of the production environment, and it has become an important production process in various enterprises. This article summarizes the research progress of continuous production methods for water-borne polyurethanes in recent decades.

### 4.1. High Internal Phase Emulsification Continuous Production Process

Dow Chemical developed the high internal phase emulsification continuous production of water-borne polyurethanes. The internal and external phases of an emulsion enter a continuous instant mixer at different flow rates and, by adjusting the ratio of flow rates, different concentrations of emulsions can be obtained. This technology is mainly based on the reactions of macromolecular polyols and isocyanates to synthesize prepolymers in which prepolymers and hydrophilic chain extenders are mixed evenly in a specific ratio to form prepolymer emulsions with a high internal phase ratio. Subsequently, it is mixed with water containing chain extenders in a specific ratio and then stirred to form a stable water-borne polyurethane dispersion [31]. Hubbard [32] uses the high internal phase ratio continuous emulsification method to produce water-borne polyurethane foam adsorbents. The prepared water-borne polyurethane adsorbents have good pore size distribution and porosity, and their adsorption performance can reach 190 g/g. This method realizes the continuous production of water-borne polyurethanes, and also greatly improves production efficiency without use of organic solvents.

### 4.2. Continuous Production by Prepolymer Process

A prepolymer containing a hydrophilic group (NCO terminal isocyanate group) is first synthesized during the synthesis of water-borne polyurethanes by prepolymer processes, which is then neutralized and dispersed. Subsequently, triethylamine is used for chain extension to obtain a stable water-borne polyurethane emulsion. The prepolymer process uses very little or even no solvent. During chain extension, multifunctional amines can be used in order to generate micro-crosslinked structures at low temperatures to improve product performance. In the 1940s, Bayer proposed the continuous synthesis of water-borne polyurethanes. In the 1950s, the reaction was used industrially. The specific reaction process involved the addition of macromolecular polyols, and hydrophilic monomers and isocyanates reacted in a pre-polymerization reactor to form prepolymers containing carboxylic acid groups. After that, the prepolymer and triethylamine were mixed in a static laminar flow mixer in order to neutralize the prepolymer. The neutralized prepolymer and aqueous solution containing small-molecule diamines were mixed in a rotor-stator mixer for the chain extension reaction to increase the molecular weight of the water-borne polyurethane. Subsequently, the solvent was recovered by continuous distillation to obtain a water-borne polyurethane with a solid content of 40% [33]. Figure 4 shows the reaction process.

### 4.3. Twin-Screw Extruder Reactor Continuous Production

The twin-screw extruder reactor with the high degree of continuity, short reaction time, good linearity of the adhesive, uniform molecular weight distribution, and high yield, which supplies a new continuous production technology that is widely used in the rubber and adhesive industries. Piqueras et al. [34] reported the solvent-free synthesis of polyurethane ionomers while using a hydrophilic monomer (2,2-dimethylolpropionic acid) in polyether and polyether polyol with a twin-screw extruder as the reaction vessel. The ionomers were dissolved in acetone, neutralized by triethylamine, dispersed in water to form an emulsion, and then the solvent was removed by distillation in order to obtain a water-borne polyurethane dispersion. In this experiment, the water-borne polyurethane dispersion that is synthesized with a carboxylate as the hydrophilic monomer had the disadvantages of a low solid content, high viscosity, and easy demulsification.

### 4.4. Continuous Production Process Using a Low-Shear Mixer

Continuous production processes while using low-shear mixers provide a facile process to synthesize polyurethane-urea dispersions. The process can be scaled-up to pilot plant scale, and eventually to commercial-scale production. Additionally, the process has a low energy consumption, high efficiency, and high yield. Aromatic diisocyanates can be used without substantial reactions between aromatic isocyanate groups and water [35].

### 4.5. Continuous Production Processes Using Spiral Dispersion

Spiral dispersion can be used for the continuous preparation of aqueous dispersions of non-self-emulsifying polymers while using an emulsifying polymer in water with the aid of a strong shear force. This method is suitable for high viscosity (initial viscosity >100 Pa·s at 45 °C) resins and it does not require a solvent. The final emulsion has fine particles and long-term storage stability. If the material exceeds the agglomeration temperature, then it must be quickly transferred to the dilution zone, and cold water should be added in order to prevent agglomeration.

### 4.6. High-Gravity Rotating Packed Bed Reactor for the Continuous Production of Water-Borne Polyurethane

Wang et al. [36] used a rotating packed bed reactor to produce water-borne polyurethane emulsions with a high solid content of 55% and low viscosity. Figure 5 shows the whole reaction process and Figure 6 shows the specific design structure and reaction mechanism of rotating packed bed. The rotating packed bed provided a strong shear force environment to break down the prepolymer into smaller particles through a wire mesh. The shear effect increased with increasing the rotation speed and gravity level, leading to further reducing the particle size. This experiment proved that the rotating packed bed emulsification method is an economical, environmentally-friendly, and low-energy consumption production strategy, with a short emulsification time and good volatility, which are conducive to commercialization and industrialization.

## 5. Applications of Water-Borne Polyurethane

As binary colloid systems, water-borne polyurethanes can meet low-carbon economy and environmental protection requirements. The hydrophilicity of the polymer is a key factor that determines the particle size range and distribution of colloidal particles. Increasing the overall content of hydrophilic groups in water-borne polyurethanes decreases the particle size and narrows the distribution, thus improving the storage stability [9]. The production process also plays an important role in improving stability. Conventional batch-fed processes produce water-borne polyurethanes and they rely on a high-speed shear stirrer, which consumes a lot of energy. It also requires long reaction time and water-borne polyurethane dispersions would not form immediately, which results in reactions between isocyanate and water, thus affecting the stability of the system and performance of the water-borne polyurethane. Continuous production processes can increase the mixing energy per unit effective volume. The prepolymers have a short residence time in water, which makes the colloidal particles evenly dispersed. The remaining isocyanate in the prepolymer reacts easily and fully with chain extenders to limit side reactions between the isocyanate and water, thus improving the performance of water-borne polyurethane.

### 5.1. Adhesives

Water-borne polyurethanes have been used as adhesives, due to their low cost, good processability, low/negligible content of volatile organic solvents, environmentally-friendly nature, and good adhesion performance for both natural and synthetic rubbers. They have been widely applied in footwear products, paper products, construction, plastic processing, and automobile decoration [37].

Daniloska et al. [38] prepared an aqueous dispersion of MoS_2_ crystals by ultrasonic exfoliation in the presence of polyvinylpyrrolidone (PVP, molecular weight = 10,000 g·mol) with MoS_2_ nanoplatelets as an enhancer. The water-based colloidal particles (latex) were synthesized by microemulsion photopolymerization in a continuous tube reactor. A polyurethane/methacrylate nanocomposite pressure-sensitive adhesive was synthesized by water dispersion blending while using a continuous process. The results showed that the nanocomposite film had good adhesion and mechanical properties upon increasing the MoS_2_ content and it was an excellent pressure-sensitive material.

### 5.2. Coatings

The environmental protection laws of various countries have imposed strict limits on the content of volatile organic compounds (VOC) and hazardous air pollutants (HAP) in paints in order to improve the quality of living environments. Hence, low-pollution paints, such as nitrocellulose and polyester paints, are greatly restricted due to the presence of volatile organic solvents [39]. Water-borne polyurethane paints use water as the dispersion medium and contain almost no organic solvents and are non-toxic, non-polluting, flame-retardant, and energy-saving [40]. The hard and soft segments in the molecular structure of water-borne polyurethanes determine the hardness and flexibility. Their two-phase structure confers water-borne polyurethanes with excellent low-temperature film-forming properties, leveling, and flexibility, good heat resistance, and stickiness [22]. Figure 7 illustrates the TEM images of WPU dispersions, homogeneous distributed spherical particle. Additionally, the hard segment content increases could affect the particle distribution of WPU dispersions [41]. Figure 8 shows the TEM micrographs of WBPU (waterborne polyurethane-fluorinated acrylic) dispersions with different PEG content. WBPU dispersions can be mixed with a polyethylene oxide (PEO) water solution to prepare a WBPU/PEO feed solution with viscosity that is suitable for three-dimensional (3D) printing [22]. Because of the presence of hydrogen bonding, they have excellent wear resistance and high hardness. They are used in continuous industrial production and have broad development prospects. Water-borne polyurethanes have been widely used in metal anti-corrosion coatings, architectural coatings, wooden furniture paints, and many other coating decoration fields, and they have some of the best overall performance.

Zhao et al. [42] synthesized a water-borne polyurethane-fluoroacrylate emulsion (WFPU) by a soap-free emulsion polymerization method while using double-bond-terminated polyurethane and short-chain fluoroacrylate (TFMEA). Aziridine cross-linking was used in order to improve the film-forming ability and mechanical performance of the fluoride-containing water-borne polyurethane. The excellent film-forming and excellent mechanical properties of WFPU allow them to be used as waterproof paints. (Figure 9) visually shows that the cotton coating hydrophobicity is effectively enhanced by WFPU. Figure 10 shows the detailed reaction mechanism.

According to Peng et al. [43], an environmentally-friendly water-borne polyurethane containing sulfonic acid groups was synthesized from an aqueous dispersion of isophorone diisocyanate, polytetramethylene glycol, and poly(1,4-butene adipate) as reagents, and aliphatic diamine sulfonate as a hydrophilic chain extender. The new sulfonic acid-based environmentally-friendly waterborne polyurethane coatings had a higher healing efficiency than coatings without sulfonic acid. The healing efficiency increased upon increasing the soft matter content. Liu et al. [44] reported that a UV-cured castor oil-based cationic water-borne polyurethane acrylate was an eco-friendly carrier for immobilized enzymes. Covalently embedding the double-bond modified lysozyme in polyurethane paints by UV curing provided a simple method for preparing highly effective sterilization surfaces. This method can be extended to other anti-pollution enzymes, such as protease, lipase, and amylase, in order to prepare highly active, stable, and environmentally-friendly antifouling paints.

Water-borne polyurethanes are widely used as finishing agents for high-end leather products. They can enable the leather to have high gloss, abrasion resistance, low-temperature resistance, abrasion resistance, good elasticity, and good hand feel, while also preventing them from breaking. Anionic water-borne polyurethanes are the main finishing agents for leather, while cationic water-borne polyurethanes are used as sealers and primers in order to give leather a soft, natural, and plump appearance [45]. They have been used as a softener and anti-wrinkle agent for fabrics by many companies and are very popular with consumers [46].

### 5.3. Sewage Disposal

Water-borne polyurethane gels have good mechanical, physical, and chemical stability, and they are easily modified by nanocomposites. They are suitable for large-scale, inexpensive processing. Water-borne polyurethanes are excellent carriers for immobilization, and they can protect and immobilize anammox bacteria and bacterial communities. This provides a promising strategy for high-nitrogen wastewater treatment. Chen et al. [47] proposed continuous-flow equipment utilizing the materials of waterborne polyurethane (WPU), sodium alginate (SA), olyvinyl alcohol (PVA), and mixed polymer PVA–SA to immobilize anammox bacteria, the reaction process illustrated in Figure 11. Moreover, Figure 12 illustrates the detailed experimental result that demonstrated WPU to have the ability to maintain a good sludge retention and excellent mechanical stability. Dong et al. [48] proposed a continuous synthesis method for water-borne polyurethane gels without crosslinkers and their application in cell embedding. During polymerization, the functional ends of isocyanate groups were blocked in order to obtain a non-toxic water-borne polyurethane.

Immobilized beads have good nitrification, and immobilized microspheres have great potential for wastewater treatment. This simple immobilization technology has promising applications for the immobilization of various microbial cells. Ya et al. investigated the effects of environmental changes, such as temperature, dissolved oxygen concentration, and pH, on the nitrification characteristics under low ammonia nitrogen conditions [49]. Specifically, suspensions and water-borne polyurethanes were used in order to immobilize nitrifying bacteria, and the results indicated that nitrification tended was complete at higher pH values, dissolved oxygen, and temperature. In other words, water-borne polyurethane immobilized nitrifying bacteria pellets showed prominent performance for treating micropollutants in water supplies.

## 6. Conclusions

Water-borne polyurethanes, as environmentally friendly functional polymers, are typically synthesized by batch-fed processes. However, batch-fed processes are limited by their poor stability, high energy consumption, hazards, and contamination risks caused by the use of acetone. When compared with conventional batch-fed processes, the continuous production of water-borne polyurethane uses water as the solvent, with a much simpler and effective process. In addition, the quantity of acetone consumed can also be reduced, even to zero in some cases. The continuous production of water-borne polyurethane can speed up the drying process and increase solid content. Indeed, water-borne polyurethanes that are produced by continuous process exhibit controllable viscosity and hydrophilicity. Hence, continuous processes are suitable for automated large-scale production and they can improve the degree of automation and product performance stability. This technique is becoming popular and it has broad application prospects.

## Figures and Tables

**Figure 1 polymers-12-02875-f001:**
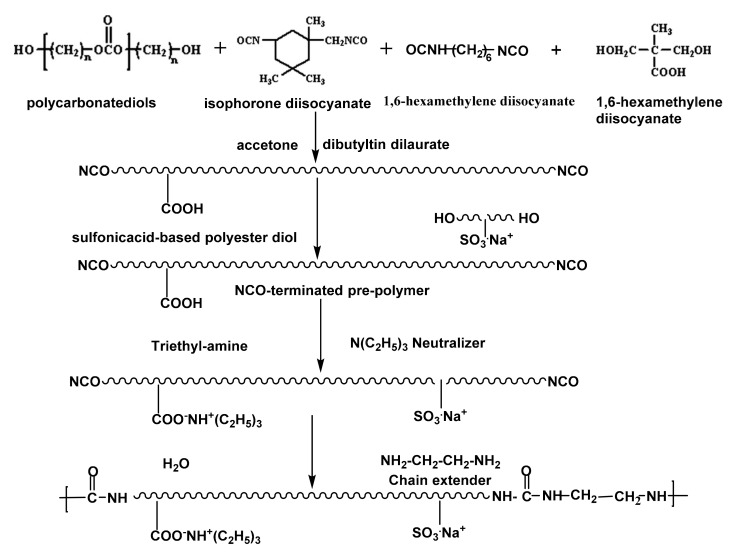
Schematic diagram of the preparation process for waterborne polyurethane dispersions [10]. Copyright © (2015) Elsevier.

**Figure 2 polymers-12-02875-f002:**
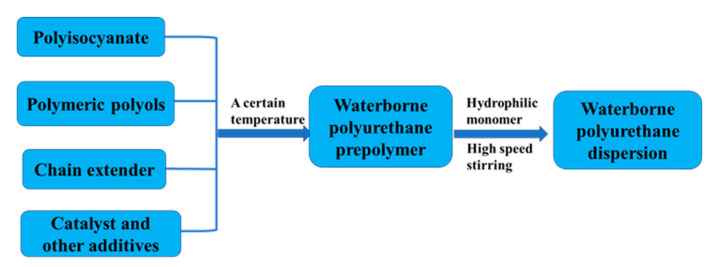
Flow chart of the reaction of polyisocyanate with polyhydroxy compound, polyether and polyester.

**Figure 3 polymers-12-02875-f003:**
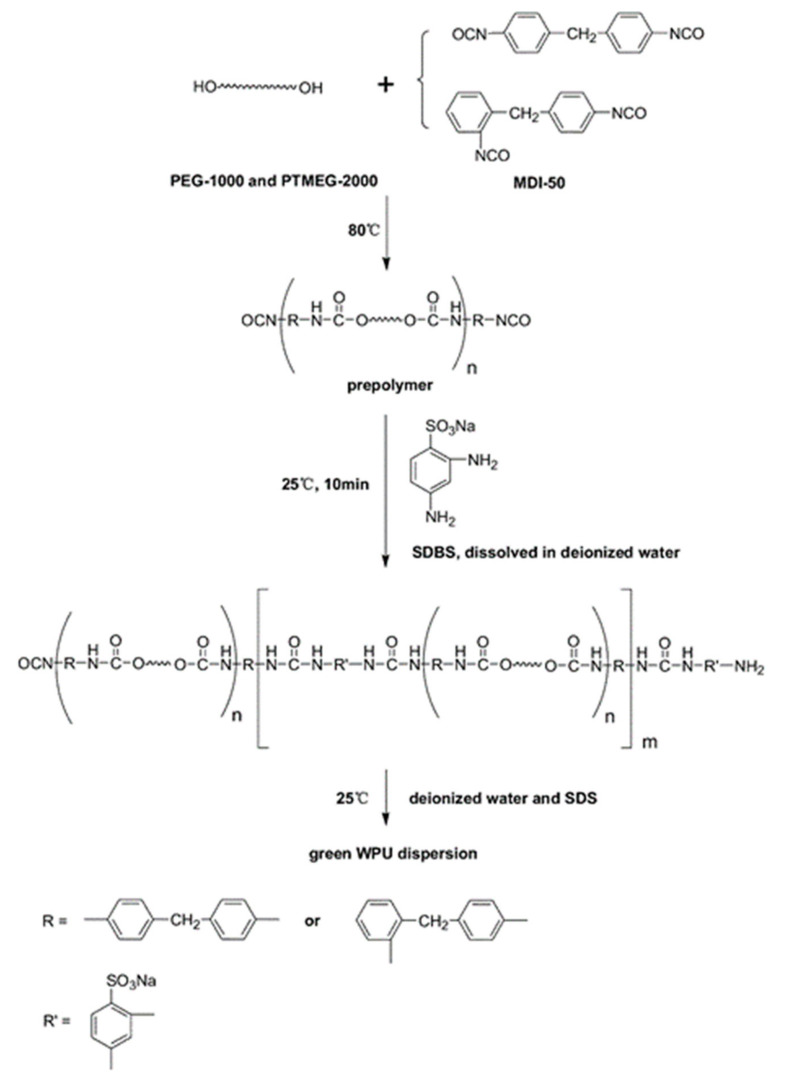
Schematic diagram of synthesis process of aqueous polyurethane dispersions [16]. Copyright © (2016) Elsevier.

**Figure 4 polymers-12-02875-f004:**
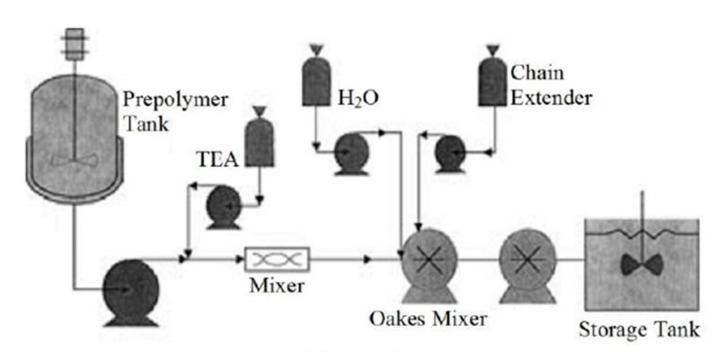
Prepolymer method continuous mixer set-up [33]. Copyright © (2010) John Wiley and Sons Inc.

**Figure 5 polymers-12-02875-f005:**
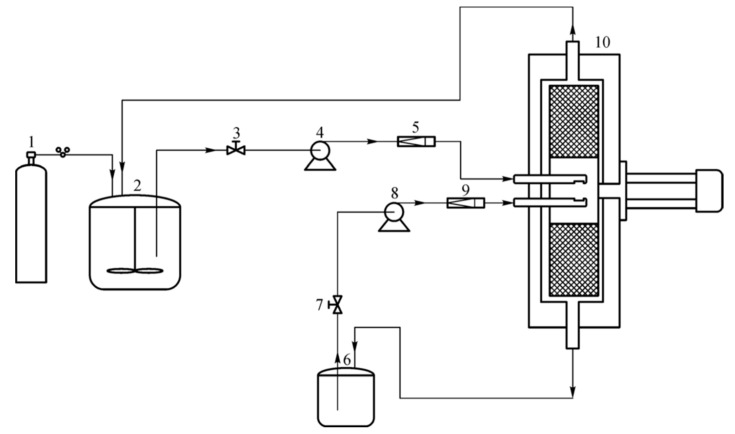
Schematic diagram of experimental setup for waterborne polyurethane (WPU) emulsification. 1: Nitrogen cylinder; 2: pre-polymerization reactor; 3, 7: valve; 4, 8: pump; 5, 9: flow meter; 6: emulsion storage tank; 10: RPB [36]. Copyright © (2020) Higher Education Press.

**Figure 6 polymers-12-02875-f006:**
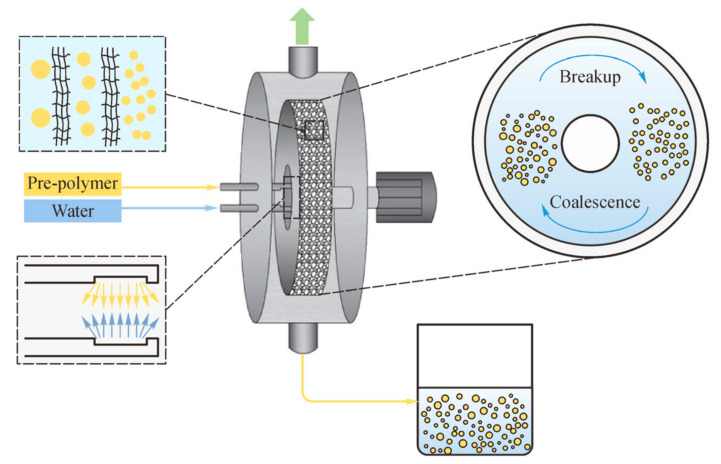
Specific design structure of the RPB reactor [36]. Copyright © (2020) Higher Education Press. Copyright © (2020) Higher Education Press.

**Figure 7 polymers-12-02875-f007:**
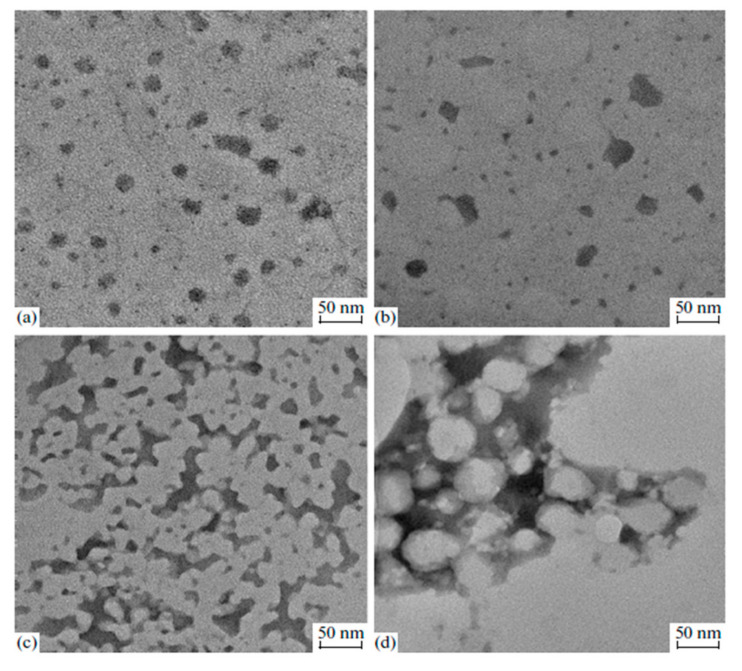
TEM micrographs of the dispersions of (**a**) PU-35, (**b**) PU-40, (**c**) PU-45, and (**d**) PU-50; (hard/soft segment molar ratio of 3.5, 4.0, 4.5, and 5.0, respectively) [41]. Copyright © (2015) Pleiades Publishing.

**Figure 8 polymers-12-02875-f008:**
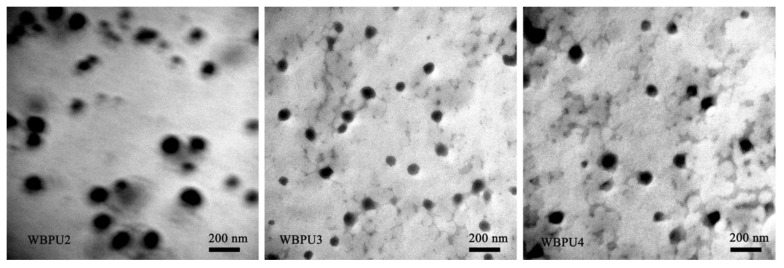
TEM micrographs of WBPU2, WBPU3, and WBPU4 dispersions [22]. Copyright © (2020) Springer Nature.

**Figure 9 polymers-12-02875-f009:**
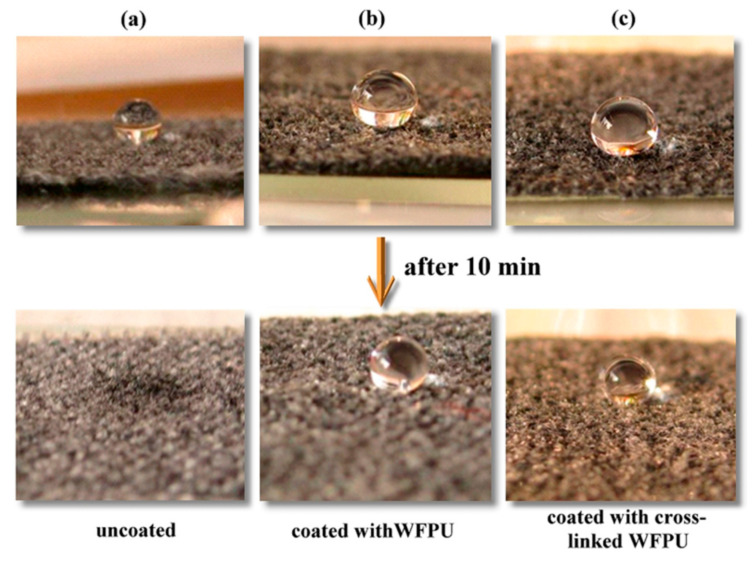
Photographs of cottons absorbing water droplets. Key: (**a**) uncoated; (**b**) coated with WFPU; and, (**c**) coated with the cross-linked WFPU [42]. Copyright © (2014) American Chemical Society.

**Figure 10 polymers-12-02875-f010:**
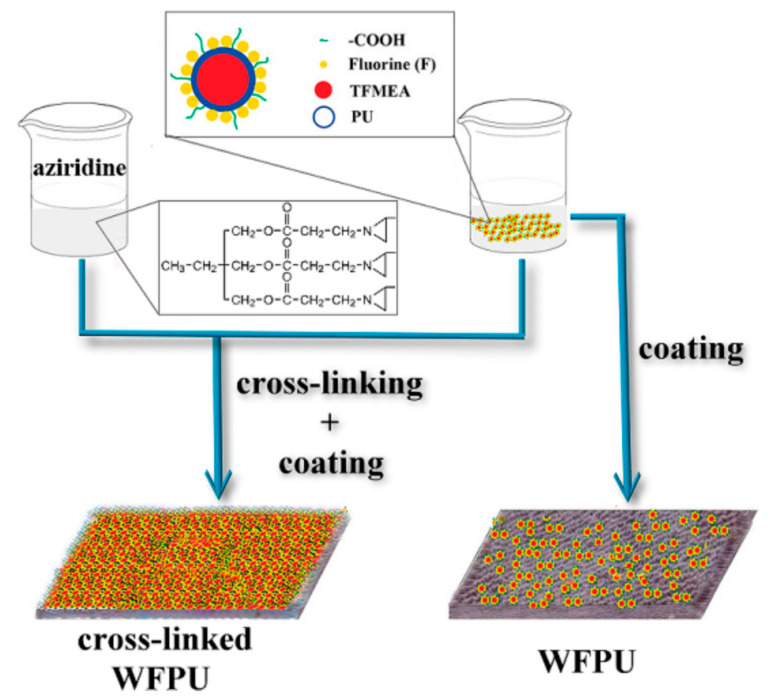
Schematic diagram of Films Preparation Process of WFPU and the Cross-Linked WFPU [42]. Copyright © (2014) American Chemical Society.

**Figure 11 polymers-12-02875-f011:**
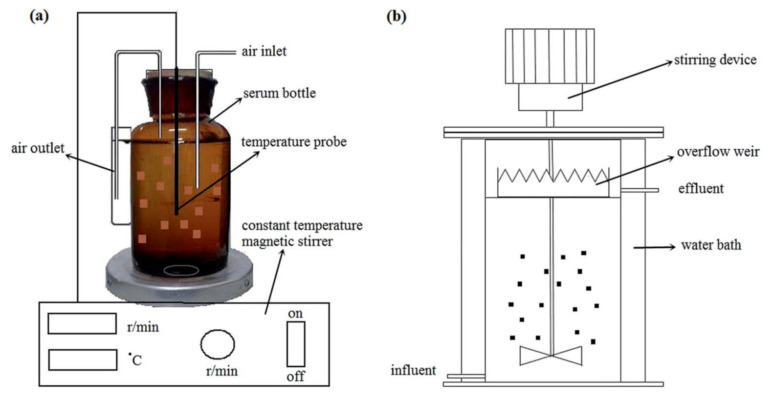
(**a**) Apparatus for measuring anammox performance. (**b**) Stirred continuous-flow reactor [47]. Copyright © (2015) Royal Society of Chemistry.

**Figure 12 polymers-12-02875-f012:**

Digital images of the 4 types of immobilized granules (**a**) PVA (**b**) SA (**c**) PVA–SA (**d**) WPU [47]. Copyright © (2015) Royal Society of Chemistry.

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
