# Peer review of "Continuous Production of Water-Borne Polyurethanes: A Review"

_polymers, 2020, doi:10.3390/polym12122875_

Round 1
Reviewer 1 Report
This manuscript focuses mainly on the continuous production process of water born polyurethanes. Here is my comments.
- English should be improved. Moreover, in the text there are erroneous repetitions of group of words that are confusing and have to be cancelled ( for instance line 49 and line 66 of p. 2, line 82 of p.3). In the abstract, please replace the two sentences in lines 12, 13 and 14 with the unique sentence: Water-borne polyurethanes have attracted increasing attention due to their extensive applications in plastics, paints, adhesives, inks, biomaterials and other fields.
- The title of paragraph 2 should be changed in “Synthesis and main properties of waterborne polyurethanes”. Paragraph 2.1 should be renamed “Structure and synthesis” and should enclose also the different classification criteria of waterborne polyurethanes.
- Reaction schemes also contain errors; for instance in scheme 1 acctone is written instead of acetone. Moreover, abbreviations (IPDI,HDI etc.) are used without any previous legend or explanation of the meaning.
- Line 139, p.5: please, eliminate “has highly unsaturated free groups and”.
- Reaction (1): please represent the mechanism with 1 molecule of isocyanate and 1 molecule of alcohol (otherwise add a second molecule of alcohol on the left side of the reaction).
- Reaction (2) does not represent a mechanism, is just an equation. Moreover the catalyst is not written in English.
- Please, check all the reference numbers in the text and figure captions. For instance, Kim et al. have been incorrectly quoted as reference 19, whereas in the references list these authors correspond to 20. Analogously, reference 16 does not correspond to Lei et al.
- Change the title of paragraph 5, for instance in “Applications of waterborne polyurethanes”.
Author Response
Dear Reviewer,
Thank you for reading our manuscript and reviewing it, which will contribute our scientific ability. I have read your comments very carefully and have made correction which we hope meet with your approval. The following details are the responses to your comments.
Point 1: English should be improved. Moreover, in the text there are erroneous repetitions of group of words that are confusing and have to be cancelled (for instance line 49 and line 66 of p. 2, line 82 of p.3). In the abstract, please replace the two sentences in lines 12, 13 and 14 with the unique sentence: Water-borne polyurethanes have attracted increasing attention due to their extensive applications in plastics, paints, adhesives, inks, biomaterials and other fields.
Response 1: Thank you for your advice. We have revised manuscript to improve the language. The details showed in manuscript, for example:
- Original line12 to line 14 “Water-borne polyurethanes have been applied in plastics, paints, adhesives, inks, biomaterials and other fileds. Water-borne polyurethanes have attracted increasing attention due to their extensive applications.” changed into “Water-borne polyurethanes have attracted increasing attention due to their extensive applications in plastics, paints, adhesives, inks, biomaterials and other fields.”
- Original line 40 to line 42 “Since their initial proposal in the 1950s, water-borne polyurethanes have been widely applied in industry and daily life due to their tunable soft and hard segments, good low-temperature resistance, high flexibility, and strong adhesion.” changed into “ It has been widely applied in industry and daily life due to their tunable soft and hard segments, good low-temperature resistance, high flexibility, and strong adhesion”
- Original line 43 to line 47 “Due to their low toxicity, inflammability, good wear and corrosion resistance, and strong adhesion, water-borne polyurethanes can be applied as construction and automotive paints, leather industry, and as adhesives. Due to their small production scale, labor-intensiveness, and the great impact of human factor on the final product, batch-fed production processes affect the quality of water-borne polyurethanes, and the resulting dispersions have a short shelf life.” changed into “Water-borne polyurethanes can be applied as construction and automotive paints, leather industry, and adhesives based on their low toxicity, inflammability, good wear and corrosion resistance, and strong adhesion. The batch-fed production process existences problems of small production scale, labor-intensiveness, and would suffer from human factors significantly, which affecting the quality of water-borne polyurethanes and resulting dispersions have a short shelf life.”
- Original line 47 to line 51 “In contrast, the continuous production of water-borne polyurethanes provides a highly-automated method that is suitable for large-scale production continuous production and the products have enhanced performance stability. In this paper, the structural characteristics and reaction mechanism, continuous synthesis methods, and the applications of water-borne polyurethanes are reviewed.” changed into “ In contrast, the continuous production of water-borne polyurethanes provides a highly-automated method suitable for large-scale production and endows the final product with enhanced performance stability. As one of the major water-borne polyurethane production technologies, continuous production technology has been involved in previous reviews [8], but has not been summarized as a special review to be best of our knowledge. This review presents recent breaking-through advance in continuously-produced water-borne polyurethane and discusses their applications in adhesive, coating and sewage disposal. This review demonstrates the effectiveness of continuous product of water-borne polyurethane and in particular, the structural characteristics and reaction mechanism, continuous synthesis methods, and the applications of water-borne polyurethanes. All of these would provide scientific researchers more convenient to research water-borne polyurethanes via continuous production.”
- Original line 62 to line 68 “Water-borne polyurethanes can be classified as cationic, anionic, zwitterionic, or nonionic, according to the ionic groups on the molecular chain and their charge. Water-borne polyurethanes are mainly divided into polyols and polyisocyanates, according to the raw materials. Polyols can be divided into polyesters, polyethers, and polyolefins, while polyisocyanates can be divided into aliphatic, aromatic, and cycloaliphatic water-borne polyurethanes. Water-borne polyurethane emulsions can be divided into externally emulsified and internally emulsified water-borne polyurethanes.” changed into“Water-borne polyurethanes can also be classified as cationic, anionic, zwitterionic, or nonionic, according to the ionic groups on the molecular chain and their charge. Moreover, water-borne polyurethanes are mainly divided into polyols and polyisocyanates, according to their reaction raw materials. For example, polyols can be divided into polyesters, polyethers, and polyolefins, while polyisocyanates can be divided into aliphatic, aromatic, and cycloaliphatic.”
Point 2: The title of paragraph 2 should be changed in “Synthesis and main properties of waterborne polyurethanes”. Paragraph 2.1 should be renamed “Structure and synthesis” and should enclose also the different classification criteria of waterborne polyurethanes.
Response 2: We have changed the title of paragraph 2 to “Synthesis and main properties of waterborne polyurethanes”. The paragraph 2.1 was renamed as “Structure and synthesis”.
Point 3: Reaction schemes also contain errors; for instance in scheme 1 acctone is written instead of acetone. Moreover, abbreviations (IPDI, HDI etc.) are used without any previous legend or explanation of the meaning.
Response 3:We have corrected the word “acctone” into “acetone” and give the abbreviations (PCDL, IPDI, HDI, DMPA, DBTDL) explanation in Figure 1.
Point 4: Line 139, p.5: please, eliminate “has highly unsaturated free groups and”.
Response 4:We have eliminated these words “has highly unsaturated free groups and” in line 139.
Point 5: Reaction (1): please represent the mechanism with 1 molecule of isocyanate and 1 molecule of alcohol (otherwise add a second molecule of alcohol on the left side of the reaction).
Response 5:We have added a second molecule of alcohol on the left side of the reaction.
Point 6: Reaction (2) does not represent a mechanism, is just an equation. Moreover the catalyst is not written in English.
Response 6: We have replaced the word “mechanism” by “equation” and revised the name of the catalyst in English.
Point 7: Please, check all the reference numbers in the text and figure captions. For instance, Kim et al. have been incorrectly quoted as reference 19, whereas in the references list these authors correspond to 20. Analogously, reference 16 does not correspond to Lei et al.
Response 7: We have checked all the reference numbers in the text and figure captions, and revised the quoted numbers of the references.
Point 8: Change the title of paragraph 5, for instance in “Applications of waterborne polyurethanes”.
Response 8: We have changed the title of paragraph 5 into “Applications of waterborne polyurethanes”.
Dear Reviewer,
Thank you for reading our manuscript and reviewing it, which will contribute our scientific ability. I have read your comments very carefully and have made correction which we hope meet with your approval. The following details are the responses to your comments.
Point 1: English should be improved. Moreover, in the text there are erroneous repetitions of group of words that are confusing and have to be cancelled (for instance line 49 and line 66 of p. 2, line 82 of p.3). In the abstract, please replace the two sentences in lines 12, 13 and 14 with the unique sentence: Water-borne polyurethanes have attracted increasing attention due to their extensive applications in plastics, paints, adhesives, inks, biomaterials and other fields.
Response 1: Thank you for your advice. We have revised manuscript to improve the language. The details showed in manuscript, for example:
- Original line12 to line 14 “Water-borne polyurethanes have been applied in plastics, paints, adhesives, inks, biomaterials and other fileds. Water-borne polyurethanes have attracted increasing attention due to their extensive applications.” changed into “Water-borne polyurethanes have attracted increasing attention due to their extensive applications in plastics, paints, adhesives, inks, biomaterials and other fields.”
- Original line 40 to line 42 “Since their initial proposal in the 1950s, water-borne polyurethanes have been widely applied in industry and daily life due to their tunable soft and hard segments, good low-temperature resistance, high flexibility, and strong adhesion.” changed into “ It has been widely applied in industry and daily life due to their tunable soft and hard segments, good low-temperature resistance, high flexibility, and strong adhesion”
- Original line 43 to line 47 “Due to their low toxicity, inflammability, good wear and corrosion resistance, and strong adhesion, water-borne polyurethanes can be applied as construction and automotive paints, leather industry, and as adhesives. Due to their small production scale, labor-intensiveness, and the great impact of human factor on the final product, batch-fed production processes affect the quality of water-borne polyurethanes, and the resulting dispersions have a short shelf life.” changed into “Water-borne polyurethanes can be applied as construction and automotive paints, leather industry, and adhesives based on their low toxicity, inflammability, good wear and corrosion resistance, and strong adhesion. The batch-fed production process existences problems of small production scale, labor-intensiveness, and would suffer from human factors significantly, which affecting the quality of water-borne polyurethanes and resulting dispersions have a short shelf life.”
- Original line 47 to line 51 “In contrast, the continuous production of water-borne polyurethanes provides a highly-automated method that is suitable for large-scale production continuous production and the products have enhanced performance stability. In this paper, the structural characteristics and reaction mechanism, continuous synthesis methods, and the applications of water-borne polyurethanes are reviewed.” changed into “ In contrast, the continuous production of water-borne polyurethanes provides a highly-automated method suitable for large-scale production and endows the final product with enhanced performance stability. As one of the major water-borne polyurethane production technologies, continuous production technology has been involved in previous reviews [8], but has not been summarized as a special review to be best of our knowledge. This review presents recent breaking-through advance in continuously-produced water-borne polyurethane and discusses their applications in adhesive, coating and sewage disposal. This review demonstrates the effectiveness of continuous product of water-borne polyurethane and in particular, the structural characteristics and reaction mechanism, continuous synthesis methods, and the applications of water-borne polyurethanes. All of these would provide scientific researchers more convenient to research water-borne polyurethanes via continuous production.”
- Original line 62 to line 68 “Water-borne polyurethanes can be classified as cationic, anionic, zwitterionic, or nonionic, according to the ionic groups on the molecular chain and their charge. Water-borne polyurethanes are mainly divided into polyols and polyisocyanates, according to the raw materials. Polyols can be divided into polyesters, polyethers, and polyolefins, while polyisocyanates can be divided into aliphatic, aromatic, and cycloaliphatic water-borne polyurethanes. Water-borne polyurethane emulsions can be divided into externally emulsified and internally emulsified water-borne polyurethanes.” changed into“Water-borne polyurethanes can also be classified as cationic, anionic, zwitterionic, or nonionic, according to the ionic groups on the molecular chain and their charge. Moreover, water-borne polyurethanes are mainly divided into polyols and polyisocyanates, according to their reaction raw materials. For example, polyols can be divided into polyesters, polyethers, and polyolefins, while polyisocyanates can be divided into aliphatic, aromatic, and cycloaliphatic.”
Point 2: The title of paragraph 2 should be changed in “Synthesis and main properties of waterborne polyurethanes”. Paragraph 2.1 should be renamed “Structure and synthesis” and should enclose also the different classification criteria of waterborne polyurethanes.
Response 2: We have changed the title of paragraph 2 to “Synthesis and main properties of waterborne polyurethanes”. The paragraph 2.1 was renamed as “Structure and synthesis”.
Point 3: Reaction schemes also contain errors; for instance in scheme 1 acctone is written instead of acetone. Moreover, abbreviations (IPDI, HDI etc.) are used without any previous legend or explanation of the meaning.
Response 3:We have corrected the word “acctone” into “acetone” and give the abbreviations (PCDL, IPDI, HDI, DMPA, DBTDL) explanation in Figure 1.
Point 4: Line 139, p.5: please, eliminate “has highly unsaturated free groups and”.
Response 4:We have eliminated these words “has highly unsaturated free groups and” in line 139.
Point 5: Reaction (1): please represent the mechanism with 1 molecule of isocyanate and 1 molecule of alcohol (otherwise add a second molecule of alcohol on the left side of the reaction).
Response 5:We have added a second molecule of alcohol on the left side of the reaction.
Point 6: Reaction (2) does not represent a mechanism, is just an equation. Moreover the catalyst is not written in English.
Response 6: We have replaced the word “mechanism” by “equation” and revised the name of the catalyst in English.
Point 7: Please, check all the reference numbers in the text and figure captions. For instance, Kim et al. have been incorrectly quoted as reference 19, whereas in the references list these authors correspond to 20. Analogously, reference 16 does not correspond to Lei et al.
Response 7: We have checked all the reference numbers in the text and figure captions, and revised the quoted numbers of the references.
Point 8: Change the title of paragraph 5, for instance in “Applications of waterborne polyurethanes”.
Response 8: We have changed the title of paragraph 5 into “Applications of waterborne polyurethanes”.
v

Reviewer 2 Report
The authors present a review on water borne polyurethanes, focusing on the characteristics of the polyurethanes and the mechanisms of their continuous production. The manuscript falls within the scope of the journal and may be considered for publication after attention to the following:
- Introduction: please state how the current review will benefit the scientific community? What knowledge gap does the review intend to fulfill?
- Line 32: change to "...hard to control performance..."
- Line 52: The heading for section 2 should be changed to reflect the contents covered. Consider deleting "performance analysis" as comparative performance analysis is not discussed in this section.
- Line 135: Please elaborate the meaning of "plastic resistance".
- Line 206: delete "increases" after "stability".
- Line 208: change "effective to improve" to "effective in improving"
- Please provide references to back claims made in lines 353 - 355.
Author Response
Dear Reviewer,
Thank you for reading our manuscript and reviewing it, which will contribute our scientific ability. I have read the comments very carefully and have made correction which we hope meet with your approval. The following details are the responses to your comments.
Point 1: Introduction: please state how the current review will benefit the scientific community? What knowledge gap does the review intend to fulfill?
Response 1: We are grateful for your suggestion. We have revised the Introduction section as follows:
We have stated the benefit in scientific community and the knowledge gap intend to fulfil. The details just showed in line 48 to line 58.
“the continuous production of water-borne polyurethanes provides a highly-automated method suitable for large-scale production and endows the final product with enhanced performance stability. As one of the major water-borne polyurethane production technologies, continuous production technology has been involved in previous reviews [8], but has not been summarized as a special review to be best of our knowledge. This review presents recent breaking-through advance in continuously-produced water-borne polyurethane and discusses their applications in adhesive, coating and sewage disposal. This review demonstrates the effectiveness of continuous product of water-borne polyurethane and in particular, the structural characteristics and reaction mechanism, continuous synthesis methods, and the applications of water-borne polyurethanes. All of these would provide scientific researchers more convenient to research water-borne polyurethanes via continuous production.”
Point 2: Line 32: change to "...hard to control performance..."
Response 2: We have revised the sentence into "...hard to control performance...".
Point 3: Line 52: The heading for section 2 should be changed to reflect the contents covered. Consider deleting "performance analysis" as comparative performance analysis is not discussed in this section.
Response 3: We have revised the heading for section 2 into“Synthesis and main properties of waterborne polyurethanes”.
Point 4: Line 135: Please elaborate the meaning of "plastic resistance".
Response 4: We have elaborated the meaning of “plastic resistance” in lines 137-138 “hydrogen bonding forms more physical crosslinks with plastic resistance.”
Point 5: Line 206: delete "increases" after "stability".
Response 5: We have deleted the word "increases" after "stability" in line 206.
Point 6: Line 208: change "effective to improve" to "effective in improving"
Response 6: We have changed the "effective to improve" into "effective in improving".
Point 7: Please provide references to back claims made in lines 353 - 355.
- Response 7: We have added the reference “K, A. B.; Kov, J. B.; Hank, M. The position of environmental protection in the value ranking of vocational education actors. Technium Social sciences Journal,2020, 11.”
